# Improving Visual Quality of Unrestricted Adversarial Examples with Wavelet-VAE

**Wenzhao Xiang** [* 1]  **Chang Liu** [* 1]  **Shibao Zheng** [1]

## Abstract

Traditional adversarial examples are typically generated by adding perturbation noise to the input image within a small matrix norm. In practice, unrestricted adversarial attack has raised great concern and presented a new threat to the AI safety. In this paper, we propose a wavelet-VAE structure to reconstruct an input image and generate adversarial examples by modifying the latent code. Different from perturbation-based attack, the modifications of the proposed method are not limited but imperceptible to human eyes. Experiments show that our method can generate high quality adversarial examples on ImageNet dataset.

## 1. Introduction

Despite the great success of deep neural networks, they have been shown to be vulnerable to adversarial examples (Biggio et al., 2013; Szegedy et al., 2013). By adding small perturbations, the well-designed adversarial examples are indistinguishable from the original ones, while the prediction labels of deep models can be confused. As the application of DNNs penetrates into various fields, The existence of adversarial examples has raised great concern about safety and robustness of DNNs.

For long, perturbation-based adversarial examples have been the focus of attention. Various adversarial attack methods have been proposed, including Fast Gradient Sign Method (FGSM) (Goodfellow et al., 2014), Projected Gradient Descent (PGD) (Madry et al., 2017), etc. However, in the actual scene, more threats to the DNNs come from the unrestricted adversarial examples. To be specific, the attacker makes large and visible modifications to the original images, which causes the model misclassification, while preserving

---
[*]Equal contribution [1]Institute of Image Communication and Network Engineering, Shanghai JiaoTong University, China. Correspondence to: Wenzhao Xiang <690295702@sjtu.edu.cn>, Chang Liu <sunrise6513@sjtu.edu.cn>.

*Accepted by the ICML 2021 workshop on A Blessing in Disguise: The Prospects and Perils of Adversarial Machine Learning.* Copyright 2021 by the author(s).

the normal observation from human perspective. The unrestricted adversarial examples put a new threat to the AI safety. Therefore, it is necessary to explore the scene and novel methods of unrestricted adversarial attacks.

Previous works have already given a formal definition of unrestricted adversarial attacks, Song (Song et al., 2018) first introduced the new threat and generates some adversarial examples from AC-GAN structure. Stutz (Stutz et al., 2019) divided the adversarial examples into two groups according to the definition of data manifold. However, limited by the model generation capability and coupling phenomenon of the latent code, most of the previous works are based on low resolution images. The image quality of the generated adversarial examples has not been taken into well consideration.

To address the aforementioned image quality problem, we introduce wavelet transform and Variational Autoencoder (VAE) (Kingma & Welling, 2013) structure to the reconstruct process. The intuition behind wavelet transform is that human eyes are not sensitive to high frequency signal changes. The function of wavelet transform is to decouple the input images into different frequency bands. Besides, to make the manipulations work in a more continuous way, we introduce the VAE structure.

In our work, we propose a novel adversarial attack algorithm based on wavelet-VAE network to encode and reconstruct the original images. By manipulating the latent space codes, we can generate adversarial examples which are imperceptible to human eyes. To be specific, we decompose the original images into different frequency bands, and a VQ-VAE (Razavi et al., 2019) network is trained to approximate the wavelet coefficients of different scales. Conditioned on the encoded latent space, we can formulate the attack problem as an optimization problem of finding a constrained latent code to maximize the adversarial loss.

Experiments on the ImageNet validation dataset (Deng et al., 2009) show that, when compared with classic perturbation-based attack methods, our method can achieve higher attack success rate with better image quality, which is measure by two metrics, FID and LPIPS.

In summary, we make the technical contributions as:

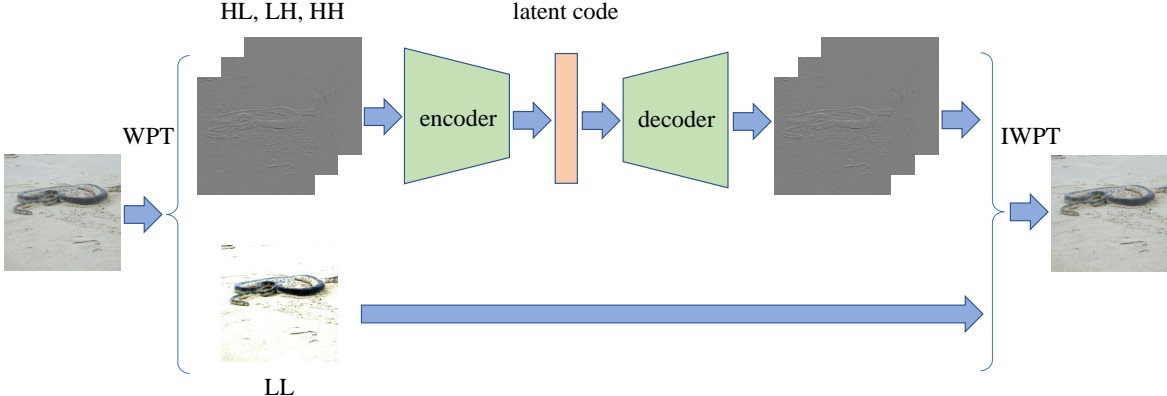

*Figure 1.* The structure of wavelet-VAE Network. The input image is first decomposed by WPT. Then the high frequency components, HL,LH and HH are sent to VQ-VAE. Finally, the reconstructed high frequency components and the original low frequency component LL are composed to the reconstructed image by inverse WPT.

- We project the real data distribution to a latent space with a wavelet-VAE network, which can decouple the human imperceptible high frequency bands by combining the wavelet transform with the Variational Autoencoder structure.

- We propose an unrestricted adversarial attack method based on the wavelet-VAE network, which is modeled as a optimization problem to the encoded latent code.

## 2. Related works

Song (Song et al., 2018) first proposed unrestricted adversarial examples and they generate new adversarial examples with AC-GAN. Kakizaki (Kakizaki & Yoshida, 2019) proposed a method to generate unrestricted adversarial examples against face recognition systems. Shamsabadi (Shamsabadi et al., 2020) proposed a black-box unrestricted adversarial attack, which modifies the color of semantic regions. Stutz (Stutz et al., 2019) provided a new way to understand the function of unrestricted adversarial attack from the manifold perspective, and the adversarial examples are generated from AC-GAN.

## 3. Method

In this section, we will describe in detail about the adversarial attack algorithm based on wavelet-VAE network.

### 3.1. Problem formulation

We start by introducing the definition of unrestricted adversarial attacks. Let $\mathcal{I}$ denotes the set of all digital images taken into consideration. The ground truth prediction can be formulated as a mapping function from the input set to the label set, i.e., $g : \mathcal{X} \subseteq \mathcal{I} \rightarrow \{1, 2, \cdots, K\}$. In addition,

a well trained classifier,denoted as $f : \mathcal{X} \rightarrow \{1, 2, \cdots, K\}$, will approximate but not equal to the ground truth function $g$. With these notations, the unrestricted adversarial examples can be defined as:

**Definition 1 (Unrestricted Adversarial Examples)**
*Unrestricted adversarial examples to a target classifier $f$ can be defined as any element in* $\mathcal{A}_u \triangleq \{x \in \mathcal{X} | f(x) \neq g(x)\}$.

The unrestricted adversarial attacks set no constraint to the modification range. However, the perception distance between the original images and the modified ones should not be too large. One possible way to achieve this goal is projecting the original high-dimensional pixel level space to a lower-dimensional manifold with VAE. Besides, another advantage of VAE is introducing noise to the latent code, which means small perturbations to the latent code will not greatly change the semantic meaning of the decoded images.

However, the VAE structure can only maintain the semantic meaning, the perceptual distance may vary greatly. Based on the fact that human eyes are not sensitive to signal changes in high frequency bands, we introduce wavelet transform to decouple the high frequency components and the low frequency ones. With the combination of wavelet transform and VAE structure, the original adversarial attack problem can be formulated as an optimization problem to the latent code of wavelet coefficients

$$\max_{\zeta} \mathcal{L}(f(\mathcal{W}^{-1}(dec(z + \zeta)))) \quad s.t. \, \|\zeta\| \leq \eta, \quad (1)$$

in which, $\mathcal{W}^{-1}$ is the inverse wavelet transform, $dec(\cdot)$ is the decoder of the VAE structure, the latent code is obtained from $z = enc(\mathcal{W}(x))$, $\zeta$ is the perturbation added to the latent code, $f$ is the target classifier, $\mathcal{L}$ is the loss function.

The adversarial examples with wavelet-VAE can be recognized as a projection from the original pixel-level manifold to the latent space of wavelet coefficients. The adversarial examples can be generated from the optimized latent code

$$x_{adv} = \mathcal{W}^{-1}(dec(z + \zeta^*)), \qquad (2)$$

in which, $\zeta^*$ is obtained from the optimal of Eq. (1).

### 3.2. Wavelet-VAE network

In this section, we will provide the detailed description about the wavelet-VAE network, which is used as the reconstruction model of the input image. The backbone network comes from VQ-VAE, which is a hierarchical learning architecture. The structure of the network is shown in Fig. 1.

We choose wavelet packet transform (WPT) to be the frequency analysis tool, which decomposes an input 2D image into four coefficients, LL (low frequency component), (HL, LH, HH) (high frequency components) , during the each decomposition level.

The wavelet coefficients obtained from WPT are sent into the VQ-VAE network. In practice, to get better image quality, we select to encode and reconstruct the high frequency components, while the low frequency components are directly sent to the inverse wavelet transform module. The overall loss function is specified in Eq. (3) as

$$
\begin{aligned}
\mathcal{L}(x, dec(e)) = & \|\mathcal{W}(x) - dec(e)\|_2^2 \\
& + \|sg[enc(\mathcal{W}(x))] - e\|_2^2 \\
& + \beta\|sg[e] - enc(\mathcal{W}(x))\|_2^2,
\end{aligned}
\qquad (3)
$$

in which, $sg(\cdot)$ refers to a stop-gradient operation that blocks gradients from flowing into its argument, $\beta$ is a hyperparameter which controls the reluctance to change the code corresponding to the encoder output, $e$ is the quantized code from $enc(\mathcal{W}(x))$. The first term in the loss function is the reconstruction loss of the selected high frequency coefficients decomposed by WPT. The second and third term are borrowed from the original VQ-VAE network, which use VQ to learn the embedding space.

### 3.3. Attack algorithm

The proposed method follows a two-stage approach: first, we train the wavelet-VAE network to encode the wavelet coefficients into a latent space, and then we add noise to the latent code to obtain an adversarial example according to the adversarial loss of the target classifier.

At the first stage, we utilize the proposed wavelet-VAE to learn the latent code distribution of the input image. At the second stage, we first fix the wavelet-VAE parameters and encode the target image to obtain the latent code $z$. Then the latent code is optimized using a gradient-based approach.

---

**Algorithm 1** Unrestricted adversarial attack based on wavelet-VAE network

1: Train wavelet-VAE to obtain the encoder $enc(\cdot)$ and decoder $dec(\cdot)$
2: **for** $x \in$ dataset **do**
3:     $z \leftarrow enc(\mathcal{W}(x))$
4:     **for** $i = 1$ to $n$ **do**
5:        $loss \leftarrow \mathcal{L}(f(\mathcal{W}^{-1}(dec(z))))$
6:        $z \leftarrow z + lr * \nabla_z loss$
7:     **end for**
8:     $x_{adv} \leftarrow \mathcal{W}^{-1}(dec(z))$
9: **end for**

---

Although the VAE structure can keep the latent code staying on the manifold, we still need to set a constraint to the $l_\infty$ norm of the latent code. The reason of doing so is that the latent code space is not strictly compact with limited dataset and VAE model. As the perturbation step gets larger, the probability of the latent code leaving the real image manifold will get higher.

## 4. Experiments

### 4.1. Experimental settings

**Dataset and Models.** The tested dataset are crafted on 1000 randomly selected ImageNet validation images. As for target models, we choose 5 state-of-the-art DNNs: VGG19 (Simonyan & Zisserman, 2014), ResNet-152 (RN152) (He et al., 2016), DenseNet-201 (Huang et al., 2017), Inception V3 (IncV3) (Szegedy et al., 2016) and Inception-ResNet V2 (IncResV2) (Szegedy et al., 2017).

**Evaluation Metrics.** In our experiments, we use Attack Success Rate (ASR) as the evaluation metrics for attack ability, which can be expressed as

$$Score_{ASR} = 100\% \times \frac{\|\{x_{adv}|f(x_{adv}) \neq y\}\|}{N}. \qquad (4)$$

The image quality is measure from two metrics fréchet inception distance (FID) (Heusel et al., 2017) and perceptual distance (LPIPS) (Zhang et al., 2018). We normalize these two metrics as

$$Score_{FID} = 100\% \times \sqrt{1 - \frac{min(FID(x, x_{adv}), 200)}{200}}, \qquad (5)$$

$$Score_{LPIPS} = 100\% \times \sum_l \frac{1}{H_l W_l} \sum_{h,w} d(f_{hw}^l, f_{0hw}^l), \qquad (6)$$

in which, $f_{hw}^l$ and $f_{0hw}^l$ are the $l$th layer feature output of VGG with the input $x$ and $x_{adv}$, $d$ is a distance metric.

**Comparison Methods.** We compared our method with the classic perturbation-based attack method, which are

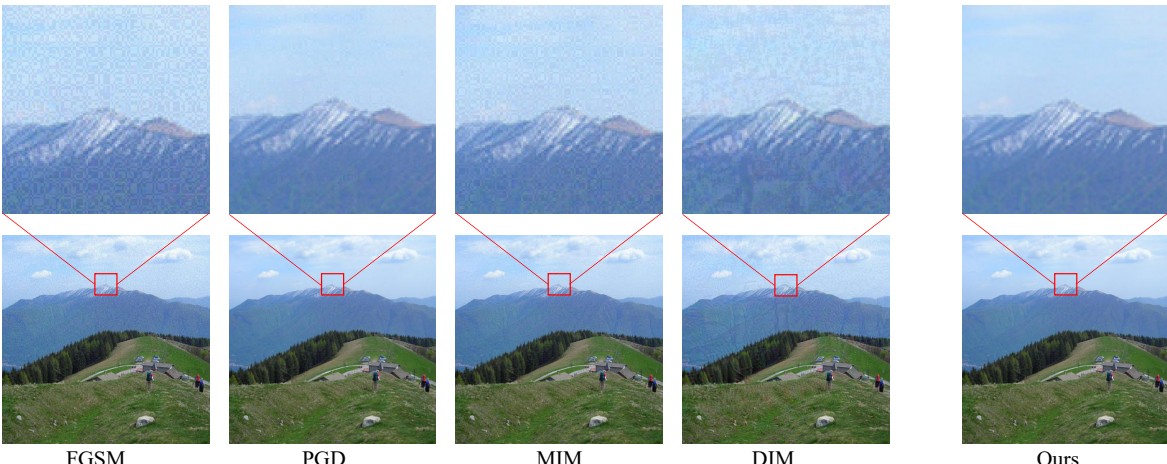

| FGSM | PGD | MIM | DIM | Ours |

*Figure 2.* The visualization results of the adversarial examples generated by different attack methods.

*Table 1.* The image quality (FID and LPIPS) of the adversarial examples generated by different attack methods on the selected models.

| Method | VGG19 | | RN152 | | DN201 | | IncV3 | | IncRes | |
|--------|-------|-------|-------|-------|-------|-------|-------|-------|-------|-------|
| | FID | LPIPS | FID | LPIPS | FID | LPIPS | FID | LPIPS | FID | LPIPS |
| FGSM | 95.20 | 87.11 | 95.44 | 87.40 | 95.36 | 88.48 | 89.96 | 84.04 | 90.47 | 83.24 |
| PGD | 99.00 | 99.74 | 99.54 | 99.95 | **99.40** | 99.93 | 97.57 | 99.92 | 97.83 | 99.96 |
| MIM | 96.85 | 94.48 | 97.08 | 94.77 | 96.92 | 94.67 | 90.57 | 92.16 | 89.34 | 91.64 |
| DIM | 86.27 | 88.34 | 85.74 | 90.18 | 85.82 | 90.22 | 83.54 | 89.01 | 83.53 | 88.34 |
| Ours | **99.65** | **99.89** | **99.70** | **99.94** | 99.33 | **99.96** | **98.98** | **99.96** | **99.96** | **99.98** |

FGSM (Goodfellow et al., 2014), PGD (Madry et al., 2017), MIM (Dong et al., 2017) and DIM (Xie et al., 2019), for both the attack success rate and the quality of the adversarial examples.

**Hyper-parameters.** We set the maximum perturbation of all the perturbation-based attack method to be $8$ with pixel value $\in [0, 255]$. For the iterative methods, the number of iteration steps is to be $100$. The decay factor of momentum is set to be $1.0$ for methods with momentum. And the transformation probability of DIM is set to be $0.7$. For our method, we set the modification constrain of the latent vector to be $\eta = 0.3$.

### 4.2. Experimental Results

We adopt a white-box attack with different attack methods on the selected classification models. And we compare the ASR and image quality of the adversarial generated by these attack methods separately. Table 1 shows the results of the image quality (FID and LPIPS score) and Table 2 shows the results of the ASR. Results from both tables show that our method can obtain higher attack success rate and higher image quality for most of the models. From Fig.2, it is shown that the adversarial examples generated by traditional perturbation-based methods have perceptible

*Table 2.* The ASR of different attack methods on the selected models.

| Attack | VGG19 | RN152 | DN201 | IncV3 | IncRes |
|--------|-------|-------|-------|-------|--------|
| FGSM | 95.8 | 88.9 | 94.1 | 78.9 | 58.9 |
| PGD | 98.2 | 98.2 | 99.4 | 97.6 | 96.0 |
| MIM | 97.8 | 98.2 | 99.4 | 97.7 | **97.0** |
| DIM | 98.2 | 98.4 | 99.5 | 97.8 | 95.7 |
| Ours | **99.9** | **98.4** | **99.6** | **99.0** | 91.0 |

noise-like patterns, while the ones generated by our method are more natural from human perspective.

## 5. Conclusion

We propose an unrestricted adversarial attack algorithm based on wavelet-VAE network. The VAE structure can make the latent code space more compact and wavelet transform can decouple the input image into different frequency bands, in which the higher frequency bands can be used to modify the image in an imperceptible way to human eyes. By adding perturbations to the latent code of high frequency wavelet coefficients, we can obtain high quality adversarial examples.

## Acknowledgements

We thank the anonymous reviewers for their valuable comments. This material is based upon CVPR-2021 AIC Phase VI Track2: Unrestricted Adversarial Attacks on ImageNet. We have won the second place in the competition.

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
