# OpenReview forum: "Improving Visual Quality of Unrestricted Adversarial Examples with Wavelet-VAE"
_ICML.cc/2021/Workshop/AML — ICML 2021 Workshop AML Poster_

### Official Review · Reviewer_wCWm · 2021-06-20
**A wavelet-VAE structure for crafting adversarial examples**

**Rating:** Accept
**Confidence:** 5

**Review:**

This paper proposes a wavelet-VAE structure to craft adversarial examples by modifying the latent code, which can generate modifications that are imperceptible to human eyes. Empirical performance can be demonstrated that the method can better mislead the classifiers by crafting adversarial examples and achieve better visual quality. However, the author is also expected to make more analysis and experiments. A clearer discussion should be considered for choosing the motivation of VAE.

---

### Decision · Program_Chairs · 2021-06-21

**Decision:**

Accept (Poster)

**Comment:**

This paper proposes a wavelet-VAE structure to craft adversarial examples by modifying the latent code, which can generate modifications that are imperceptible to human eyes. The authors can further address the reviewer's comments.